# An Automated High-Accuracy Detection Scheme for Myocardial Ischemia Based on Multi-Lead Long-Interval ECG and Choi-Williams Time-Frequency Analysis Incorporating a Multi-Class SVM Classifier

**DOI:** 10.3390/s21072311

**Published:** 2021-03-26

**Authors:** Ahmed Faeq Hussein, Shaiful Jahari Hashim, Fakhrul Zaman Rokhani, Wan Azizun Wan Adnan

**Affiliations:** 1Biomedical Engineering Department, Faculty of Engineering, Al-Nahrain University, Baghdad 10072, Iraq; ahmed.f.h.1976@gmail.com; 2Department of Computer & Communication Systems Engineering, Faculty of Engineering, Universiti Putra Malaysia, Serdang 43400, Malaysia; fzr@upm.edu.my (F.Z.R.); wawa@upm.edu.my (W.A.W.A.)

**Keywords:** CVD, Choi-Williams distribution, multi-class SVM, myocardial infarction (MI) detection, automated heart disease detection, medical screening, ECG

## Abstract

Cardiovascular Disease (CVD) is a primary cause of heart problems such as angina and myocardial ischemia. The detection of the stage of CVD is vital for the prevention of medical complications related to the heart, as they can lead to heart muscle death (known as myocardial infarction). The electrocardiogram (ECG) reflects these cardiac condition changes as electrical signals. However, an accurate interpretation of these waveforms still calls for the expertise of an experienced cardiologist. Several algorithms have been developed to overcome issues in this area. In this study, a new scheme for myocardial ischemia detection with multi-lead long-interval ECG is proposed. This scheme involves an observation of the changes in ischemic-related ECG components (ST segment and PR segment) by way of the Choi-Williams time-frequency distribution to extract ST and PR features. These extracted features are mapped to a multi-class SVM classifier for training in the detection of unknown conditions to determine if they are normal or ischemic. The use of multi-lead ECG for classification and 1 min intervals instead of beats or frames contributes to improved detection performance. The classification process uses the data of 92 normal and 266 patients from four different databases. The proposed scheme delivered an overall result with 99.09% accuracy, 99.49% sensitivity, and 98.44% specificity. The high degree of classification accuracy for the different and unknown data sources used in this study reflects the flexibility, validity, and reliability of this proposed scheme. Additionally, this scheme can assist cardiologists in detecting signal abnormality with robustness and precision, and can even be used for home screening systems to provide rapid evaluation in emergency cases.

## 1. Introduction

Cardiovascular diseases (CVD) can be classified as the most common cause of death in the world. The WHO (World Health Organization) reported that in 2014, more than 17.5 million deaths were due to CVD. This represents 31% of total global deaths. This death toll is expected to increase to approximately 24 million by 2030 [1]. One of the main forms of CVD is myocardial ischemia, which can be defined as heart muscle failure, due to insufficient blood supply. This renders CVD management and detection essential tasks for the advanced prevention, recognition, and treatment of CVDs [2]. During 2019, ischemic heart disease led to 8.9 million deaths [2]. Atherosclerosis, the primary cause of CVD, is caused by the development of fibrous plaques on the inner artery walls as thick layers of cholesterol tissue which obstruct normal blood flow. As a result, the heart muscle is deprived of the required amount of blood. With acute myocardial infarction, the dead cardiac muscles lose their contractibility at the region of ischemia [3,4]. Contemporary studies on heart disease point to the essentiality of an early CVD detection system [5,6]. The electrocardiogram (ECG) represents the preferred and most popular diagnostic approach for heart disease interpretation. This is attributed to its low cost, reduced side effects, and non-invasive test capability. The electrocardiogram reflects the heart’s electrical activity and provides important indications concerning its rhythm. The ECG composite from P, QRS, and T waves and the morphological alterations in these waves point to the heart events of arrhythmia, infarction, or ischemia [7,8]. In normal ECG cases, the electrical potential of the ST and PR segments are almost identical. However, with the presence of ischemia, the ST wave level rises or dips in accordance with the possible PR wave level [9,10]. Ischemia causes an ST normal-level deviation or PR morphological change. The PR segment provides a form of voltage reference level for the ST segment. Therefore, the PR segment may vary regardless of the ST segment, and this appears as a ST segment modification. For the most part, the ST wave deviation is due to injury to the myocardial cells. However, some myocytes are influenced to be unresponsive to repolarization, which leads to voltage disturbance, and this appears as ST level deviation [11,12]. Figure 1 shows a normal ECG case and an ischemic ECG case.

Many ischemic classification methods have been developed over recent years. Various signal-processing techniques are employed to achieve different processing algorithms. These algorithms extract the necessary features from the ECG signal to classify them as normal or abnormal cases through established processing methods which include discrete wavelet transform (DWT), time analysis, frequency analysis, time-frequency distribution (TFD), and tuneable Q wavelet transform (TQWT). Most ECG classification and diagnostic algorithms focus on the feature extraction of heart rate variability (HRV); beat-based techniques such as those described by M. Arif et al. [9], L. Sun et al. [13], N. Safdarian et al. [14], A. Kaveh et al. [15],U.R. Acharya et al. [3], B. Liu et al. [16], Murthy et al. [17], and L.N. Sharma et al. [18]; or frame-based techniques (which consider a few successive beats for short periods, usually up to 5 s), such as those described by CC Hsu [19], Jen Hong Tan et al. [20], U. Rajendra Acharya et al. [21], and Vidya K. Sudarshan et al. [22]. Of late, however, several studies have shown that the heart problem detection rate can be improved by analysing a long series of beats (more than a few seconds) [23]. This is a significant revelation in terms of early heart disease detection, where the capacity to swiftly identify and recover from abnormal heart activity is vital. An extended ECG duration process is essential for enhancing both the total scheme efficiency and overall accuracy.

In this paper, a new automated detection scheme for myocardial ischemia is presented. This method entails the extraction of a power spectrum feature component related to the ST and PR segments from the ECG signal by measuring the mean value of Choi-Williams time-frequency distribution (CWD)-related components. This scheme segments the ECG signal into 1 min time slots for extended and comprehensive spectrum analysis to indicate any pre-change in the potential MI patient. The indicator factor (F), which carries the ECG signal identification, is calculated by feeding the control ECG database for both normal individuals and patients as well as the ECG signal to be tested to the multi-class support vector machine (multi-class SVM) classifier. The usage of multi-lead ECG in this scheme provides greater accuracy and sensitivity. This new concept involves the use of a series of ECG beats for 1 min to determine whether or not the condition of a potential patient requires monitoring. This feature renders the screening and early detection process easier for home users, as it removes the need for any significant experience or skill. As shown in Figure 2, the proposed scheme delivers a high classification accuracy of 99.09%.

## 2. Methodology

### 2.1. Data Sources

Three ECG databases were used to validate the method proposed through this study. The main source of these databases is the open access Physionet database website. For normal cases, we used the MIT Fantasia Normal database as control datasets to train the multi-class SVM. As for the patient cases, we used the European ST-T database [25]. For further validation, besides the MIT Physionet database we also collected ECG data for 30 ischemia patients from the coronary care unit (CCU) in IBN-AL-NAFEES specialist heart disease hospital in Baghdad, Iraq. Table 1 provides a summary of the databases used for this study. For this undertaking, 80% of the contents from the datasets illustrated in Table 1 were used for the training process, 10% for the validation of the proposed scheme, and 10% for the testing process. As is presented, the proposed method is based on using long intervals or segments (1 min) and the training, validating, and testing ratios are calculated according to the total dataset length for each lead.

### 2.2. ECG Data Pre-Processing

The ECG data utilized in this study were subjected to pre-processing before being analysed through the proposed scheme. As the sampling rate of each database is dependent upon different parameters, a re-sampling process was conducted to render the data uniform prior to processing. We opted for the maximum sampling rate (250 samples per second) and the up-sampling of all the databases used.

Digital filters were used to remove the artifacts on the ECG, which affects the quality of the collected ECG, while a 0.5 Hz high-pass Finite Impulse Response (FIR) filter was used to remove the baseline wander shifting noise. Additionally, a bandpass FIR filter at a bandwidth of 2 Hz and 35 Hz centre frequency, was applied to eliminate muscle noise artifacts [26,27,28]. The ECG raw data were then scaled to eliminate dynamic interim changes in the ECG signal, stemming from human physiological subjects and activities. Scaling is essential for unifying the comparison among various ECG databases and feature extraction into one general algorithm.

### 2.3. Choi-Williams Time-Frequency Distribution (CWD)

The main objective of time-frequency analysis is to formulate a function which can describe the signal energy density simultaneously in both time and frequency. This can be manipulated for use in a wide range of applications that have a time-varying spectrum [23]. The use of time-frequency distribution is crucial for non-stationary and multi-component ECG signals [29]. There is no fixed representation for signal energy distribution over time and frequency, and this has led to several proposals for determining them [30]. One of the most significant and well-known representations of signal energy distribution is the quadratic class, or Cohen’s class. However, Cohen’s class distribution is hampered by unwanted cross-terms between frequency components. Among the best methods for reducing these terms to the minimum amount is the Choi-Williams class kernel CWD (t,f) [31]:(1)$$CWD\;\left(t,\omega \right)=\;\int e^{-i\omega \tau }\int \sqrt{\frac{\sigma }{4\pi \tau^{2}}}e^{-\frac{\sigma {\left(\mu -t\right)}^{2}}{4\tau^{2}}}\;x\left(\mu +\frac{\tau }{2}\right)x^{*}\left(\mu -\frac{\tau }{2}\right)\;d\mu \;d\tau ,$$
where *σ* is the real parameter, which can control the resolution as well as reduce the cross-terms.

The discrete form of CWD for input ECG sampled data *x*(*n*) can be expressed as follows [32]:(2)$$CWD\left(n,\omega \right)=\;2\;\sum_{k=-N/2}^{N/2-1}e^{-i2\omega k}\sum_{m=-M/2}^{M/2}\sqrt{\frac{\sigma }{4\pi k^{2}}}e^{-\frac{\sigma {\left(m-n\right)}^{2}}{4k^{2}}}\;x\left(m+k\right)x^{*}\left(m-k\right).$$

Pereira [33] proposed more cross-term reduction for Cohen’s class so that only the positive matrix part contains the data information, while the negative part is related to the cross-terms and can be excluded from the Choi-Williams matrix result. This concept has been extended to various Cohen’s class applications, such as electromyogram (EMG) signal and ultrasound signal.

The signals have time-dependent frequency components, and the instantaneous spectrum contains the power density attribute or marginal conditions, so that [34]:

The integration of time-frequency distribution with respect to frequency is instantaneous power:(3)$$\int CWD\left(t,\omega \right)\;d\omega =\;{\left|s\left(t\right)\right|}^{2}.$$

Moreover, the integration of time-frequency distribution with respect to time is the energy spectrum:(4)$$\int CWD\left(t,\omega \right)\;dt=\;{\left|S\left(\omega \right)\right|}^{2},$$
while the total power of distribution is given as:(5)$$E_{T}\left(CDW\left(t,\omega \right)\right)=\;\iint CDW\left(t,\omega \right)\;d\omega \;dt.$$

Many types of Cohen’s class-based distributions are employed to analyse the biosignal. As for ECG analysis, the Choi-Williams distribution delivers reliable and highly accurate results with the capacity to find the middle ground between unwanted cross-terms and the intensity of the Time-Frequency Distribution (TFD) [35].

The output of time-frequency distribution, the various components of joint time and frequency planes, as well as the effect of cross-terms are portrayed in Figure 3.

The various frequency components of the ECG signal can be picked up directly from the TFD plane. Figure 4 shows the regions of features (ST and PR segments) and noise (i.e., 50 Hz voltage mains noise. for example). The selection of these regions is dependent on the related boundary (start and end) points from the frequency axis (X axis).

### 2.4. The Multi-Class SVM Classifier

The support vector machine (SVM) technique, which many consider superior to other classification approaches, is used to determine if ECG input data are normal or an indication of ischemia [21,36]. Developed by Vapink, SVM is a two-class (binary) supervised learning technique. For this study, the use of a classifier with more than two classes is essential, as the scheme design calls for a classifier with three classes for the classification process. The multi-class SVM procedure can be carried out using a multiple trained binary SVM [29,37]. The SVM approach is based on the structural risk minimization (SRM) concept, which entails a reduction in the error rate for the outside training data set. In the SVM technique, the non-linear classification problem or the input vector classification problem are solved by mapping the high-resolution feature space through non-leaner mapping (linear classification). Consider a training dataset D of n points in the form D=xi,yii=1n, where the input data vector is xi∈ ℝP and yi∈ −1,+1 indicates the class data label of the input xi. The SVM model for the two-state classification (+ve or −ve) is a quadratic programming issue expressed as [38]:(6)minw,a,ξJw,a,ξ = 12wTw+ γ∑i=1nξi,
(7)yiwTφxi + a ≥1−ξi, ξi≥0, i = 1,2,…,n,
where:

*w* is a normal hyperplane vector,

*a* is the bias,

$\varphi \left(x_{i}\right)\;$ is a non-linear function which directs *x* to a high-dimensional space,

$\xi_{i}$ is the misclassification error,

γ is a constant which provides a trade-off control between classification margins and classification errors.

The optimized expressions of Equations (6) and (7) are:(8)$$\underset{\alpha }{\max}\sum_{i=1}^{n}\alpha_{i}-\frac{1}{2}\sum_{i,j=1}^{n}\alpha_{i}\alpha_{j}y_{i}y_{j}K\left(x_{i},x_{j}\right),$$
and
(9)$$\sum_{i-1}^{n}\alpha_{i}y_{i}=0,0\leq \alpha_{i}\leq \gamma ,i=1,2,\;\ldots ,n,$$
where:

$\alpha_{i}$ are the Lagrange multipliers,

$K\left(x_{i},x_{j}\right)$ is a kernel function that can be expressed as follows:(10)$$K\left(x_{i},x_{j}\right)=\;\varphi {\left(x_{i}\right)}^{T}\varphi \left(x_{j}\right).$$

The sequential minimum optimization algorithm can be used to solve Equations (8)–(10). The class label prediction of a testing vector *x* is expressed as follows:(11)$$y\left(x\right)=\;\left\{\begin{array}{c} +1,\;\sum_{j=1}^{n}\alpha_{j}y_{j}K\left(x,x_{j}\right)\;+\;c\;\geq 0\\ -1,\;\sum_{j=1}^{n}\alpha_{j}y_{j}K\left(x,x_{j}\right)\;+\;c<0 \end{array}\right..$$

In a multi-class SVM, the combined optimization issue can be solved by extending the binary SVM to a multi-class SVM [39]. The solution of the optimization problem leads to the definition of K hyperplanes by *W_k_*, where *k* = 1, 2, …, and *K* has the form of:(12)$$\underset{w_{k},\;a_{k}}{\min}\sum_{k=1}^{K}\frac{1}{2}w_{k}^{T}w_{k}+C\sum_{i=1}^{N}\sum_{k\;\neq \;l_{i}}\xi_{i}^{k},$$
constrained by:(13)$$w_{l_{i}}^{T}x_{i}+a_{l_{i}}\geq \;w_{k}^{T}x_{i}+b_{k}+2-\xi_{i}^{k},\;\xi_{i}^{k}\geq 0,\;i=1,2,\;\ldots ,\;N\;,\;k\neq l_{i}.$$

Equation (12) formulates the optimization and obtains the following:(14)$$\underset{a_{k}}{\max}\sum_{k=1}^{K}\sum_{i=1}^{N}\sum_{j=1}^{N}\left(a_{i}^{k}a_{j}^{l_{i}}-\frac{1}{2}a_{i}^{k}a_{j}^{k}-\frac{1}{2}a_{i}a_{j}c_{j}^{l_{i}}\right)x_{i}^{T}x_{j}+2\sum_{k=1}^{K}\sum_{i=1}^{N}a_{i}^{K},$$
where:(15)$$\sum_{i=1}^{N}a_{i}^{k}=\sum_{i=1}^{N}c_{i}^{k}a_{i},k=1,\;2,\;\ldots ,\;K,\;0\leq a_{i}^{k}\leq c,a_{i}^{l_{i}}=0,i=1,2,\;\ldots ,N,\;k\neq l_{i}.$$

As can be gathered from Equation (14), the binary (two-class) SVM can be determined from the multi-class SVM at *k* = 2 [40,41]. The classifier is employed to train the extracted features, which are used to form the proposed indicator factor (F).

The extracted features from different ECG signal leads are fed to a multi-class SVM classifier for acquiring a high classification accuracy, sensitivity, and specificity. The ST segment and PR segment features are mapped to a multi-class SVM area for the x-axis and y-axis consequently. These processes are repeated for all available ECG channels being used. For the SVM setup, the Gaussian kernel is selected for calculating the required boundaries among SVM classes where this kernel can give better results [42].

For code design selection to compute multi-class non-linear SVM, the one-against-one method is selected. Despite the fact that the one-against-all method requires fewer computational processes (number of learn), where it takes *K* vs. *K*(*K* − 1)/2 for one-against-one, the one-against-one can give a better classification accuracy depending on the intended use [36].

### 2.5. Classifier Structure

The use of multi-class SVM is essential to simulate the actual physician’s behaviour when starting to check the ECG of the potential patient. The use of many leads can give a more complete vision regarding the heart situation, which leads to accurate interpretation.

A total of two feature components are evaluated for each class using the most prevalent classifiers—support vector machines (SVMs) for ECG arrhythmia classification methods.

A hyperplane is to be found by the SVM algorithm and the distance from the hyperplane to the nearest training samples is the greatest. In other words, the optimal segmentation hyperplane maximizes the boundary of the training sample. The optimal separating hyperplane is represented by a decision function that is learned in the subsequent tests from the training set to predict the class label. We assume in our research that a training set consists of *M* samples, {(*x_i_,y_i_*), *i* = 1, …, *M*}, where *x_i_* is the vector of feature of the *i*th elements and *y_i_* is the corresponding class label. The decision-making can be expressed as:(16)$$R=\sum_{i\in SVs}^{}W_{i}*K\left(x,x_{i}\right)\;+\;b,$$
where $K\left(x,x_{i}\right)$ is the kernel function. By mapping data to high-dimensional space, the kernel function can solve the problem of linear indivisibility in primitive space. In this study, the radial basis function (RBF) is employed to apply the SVM method for the classification of ECG signals. The RBF can give a higher accuracy than other functions [43]. The RBF can be defined as follows:(17)$$K\left(x,x_{i}\right)=exp\left(\frac{{\left(x-x_{i}\right)}^{2}}{2\sigma^{2}}\right).$$

The corresponding training data are termed as support vectors (*SVs*)—that is, ∈*SVs.* The parameter *σ* affects the precision of the SVM classifier, since it can control the SVM margins. According to Sai Li et al. [44], the optimal value of *σ* in such an application should be 0.1.

### 2.6. Algorithm

The ECG signal reflects the heart’s electrical activity, which approximately portrays the general condition of the heart. The ECG composite waves P, QRS, and T describe the heart ventricle and atrium action, with other muscle part combinations during blood circulation cycles. The influence of myocardial ischemia heart disease more affects the ST segment of the ECG signal than the PR segment in the frequency ranges of 11–14 and 5–8.5 Hz, respectively [45,46].

The ST and PR segments are fed to the SVM hyperplane (Equation (11)) for the development of the classification domain. The resultant tangible values of these segments are used in a manipulated domain of an SVM multiclass classifier for each lead or ECG channel. The ST-related features are used to identify the x-axis, while the PR-related features are used to identify the y-axis of the classifier domain.

Our proposed algorithm is based on the design of an indicator factor (F) that contains the power estimation components for ECG signal parts specified by ST and PR frequency bands for each one-minute segmented slot in each ECG record. We used the PTB database as control data to determine the (F) values [47]. The PTB database holds both normal and myocardial infarction cases, which represent the advanced myocardial ischemia stages. The power estimation process begins with the computation of the CWD components with regard to time as well as frequency directions for the input data [42,48]. Subsequently, the power estimation is calculated by considering the mean value of the time-frequency output matrix. The value of I for different ECG leads varies according to various wave [49,50] components, which change from one lead to another. This provides an indication of the heart condition, which is used for the classification process. Each ECG lead is run individually, followed by cross-matching, to raise the level of accuracy and reduce the possibility of errors. The algorithm is summarized as follows:

Part one: indicator factor (F) computing.

Step 1: Pre-processing of raw ECG data from the PTB database. During this pre-processing exercise, raw ECG data are prepared for analysis. The ECG data are re-sampled, filtered, and scaled to remove unwanted artifacts, which may affect the classification results.Step 2: The output data from step 1 are segmented into 1 min slots. This segmentation is performed for all ECG data used in this study.Step 3: The ECG 1 min data are decomposed with the use of CWD in the time and frequency slots. Each slot contains the relative components which describe the instantaneous heart activity condition according to its electrical changes. The CWD is applied to each ECG lead individually.Step 4: The negative elements are omitted from the CWD main matrix. If is the CWD matrix element, then:

(18)$$CWD{\left(t,\omega \right)}_{p}=\left\{\begin{array}{c} CWD\left(t,\omega \right),\;\;\;\;\;\;\;\;E_{i}>0\\ 0\;\;\;\;\;\;\;\;\;\;\;\;\;\;\;\;\;\;\;,\;\;\;\;\;\;\;E_{i}\;\leq 0 \end{array}\right..$$

Step 5: Feature extractions are performed by estimating the CWD ST and PR components using the mean values of certain frequency bands, as portrayed below:

(19)$${\hat{P}}_{ST}\left(t,\omega \right)=Mean\left(\sum_{11}^{14}\sum_{0}^{n}CWD{\left(t,\omega \right)}_{p}^{num}\right)\;\mathrm{Hz},$$

(20)$${\hat{P}}_{PR}\left(t,\omega \right)=Mean\left(\sum_{5}^{8.5}\sum_{0}^{n}CWD{\left(t,\omega \right)}_{p}^{num}\right)\;\mathrm{Hz},$$

where:

*num* is the total CWD bins for time and frequency,

*n* is the CWD time slot.

Step 6: The indicator factor (F) is computed by mapping the results from (5) into the multi-class SVM classifier for potential patients. This facilitates the identification of learned margins for the classification of unknown input data.

Part two: normal operation.

Step 1: Repeat the process of part one from step 1 to step 5, except for the use of input data for testing instead of training data.Step 2: The result from step 5 is fed into the ex-trained multi-class SVM.Step 3: The data are classified according to decision-making in Equation (16).

## 3. Results

For this undertaking, four different databases were used to demonstrate the proposed scheme for myocardial ischemia detection. A total of 2195 min of control data for both normal and myocardial infarction cases, 4800 min for normal cases, and 21,960 min for myocardial ischemia cases were tested and evaluated from 92 normal, 148 myocardial infarction, and 109 myocardial ischemia persons.

### 3.1. The Choi-Williams Feature Extraction Process

As mentioned earlier, in this study the required features were extracted using CW-TFD. The ST and PR segment regions were used to distinguish the MI effects in the ECG recorded parts. The result from the CWD matrix is processed for each ST and PR, as stated in the previous section. The result derived is for a 1 min input ECG interval, whereby the MI changes can be captured. At the close of this process, the tangible values are fed into the next step.

### 3.2. Indicator Factor (F) Calculation

The features extracted from different ECG signal leads are fed into the multi-class SVM classifier to realize a high classification accuracy and sensitivity. The features of the ST and PR segments are mapped to the multi-class SVM area for the x-axis and y-axis, respectively. These processes are repeated for all the available ECG leads. The indicator factor F is determined by extracting the ST and PR features from the PTB database records, with a 1 min interval for each, followed by the identification of the optimal boundary for the normal region and ischemic region through the SVM classifier. Figure 5 shows the calculation for different corresponding leads for F, while Figure 6 and Figure 7 show the ST and PR boundaries, calculated by way of the SVM classifier. Figure 8 provides the results from the testing process to identify the MI cases. The combination selection is based on the databases used in this study. For each case, we used the same lead for both normal and ischemic cases to generate the learning datasets. The generated datasets were later used to distinguish the unknown ECG signals. The ST value and the PR value are a normalized percentage of power component estimation derived from the heart contraction and diastole cycles. Multi-lead feature extraction is imperative, as CVD is frequently undetectable with a single lead. The results for the accuracy, sensitivity, and specificity of the trained classifier datasets are exhibited in Table 2.

### 3.3. Classified Unknown Input Data

The automated classifier is applicable, following the training process with well-interpreted ECG data, for various leads. The ECG data stream is fed into the classifier for each particular lead. To ensure the efficiency of the scheme, various ECG datasets are used as unknown cases for analysis. Appendix A
Appendix A shows 88 records for 78 ischemic patients, available in the European ST-T Database, from the MIT Physionet website. This database contains 120 min of ECG recording for two different leads in each record. However, these leads are not the same for all patients. For example, patient e0103 has leads V4 and III, patient e0110 has leads V3 and III, and patient e0817 has leads V5 and V1, etc. To counter this impediment, we used different leads for training. Appendix A shows the analysis result for a normal case. Appendix A displays the classified results for data collected from IBN-AL-NAFEES Hospital in Baghdad, Iraq. These data were gathered from 30 myocardial ischemia-afflicted patients (subjected to constant monitoring) warded at CCU units in a hospital specializing in heart surgery. The raw data from this hospital were gathered using seven (7) ECG leads, which were arranged in separate databases. Figure 6 shows the classification results for the different databases used in this study.

## 4. Discussion

A new scheme for automated myocardial ischemia detection was proposed in this study. Four different normal and myocardial ischemia databases were used to train, validate, and test the proposed scheme. This undertaking involved an investigation of 92 normal and 266 MI subjects. Generally, the use of ST and PR extracted features yielded a higher detection performance when compared to other detection methods in this field. Based on the results exhibited in Appendix A, it can be surmised that the proposed scheme is significantly efficient, robust, and reliable for use as a preliminary indication tool for MI interpretation. Additionally, the combination of multiple extracted features into a single structure renders the proposed scheme quick and less time consuming. A 10-fold cross validation exercise was conducted to authenticate the effectiveness of the SVM performance.

With the Pereira process, the use of the Choi-Williams distribution led to a decrease in the number of cross-over frequency components and an improvement in the overall accuracy. In addition, time-frequency distributions, with regard to time, provide instant frequency information. This is useful for catching a rapid signal shift linked to the illness that triggered it. Generally, there is more memory space available for the time-frequency distribution process. With the use of the optimized and effective memory model developed by J.M Toole O’ [51], this situation can be controlled. The use of this memory model often means that the cost of computing the proposed scheme is relatively low.

For our proposed scheme, the employment of a multi-class SVM is indispensable. The classification process necessitates the inclusion of three classes for the identification of the unknown input ECG data. With this scheme, the selection of related MI ECG features (ST and PR segments) simplifies the classification process by excluding the need for any supplementary processes.

Table 2 and Figure 8 summarize the overall results with regard to the accuracy, sensitivity, and specificity of the proposed scheme for the different databases used for this investigation. The high-accuracy results indicate that the proposed scheme interprets the unknown cases efficiently. This renders it applicable in various settings (such as hospitals, clinics, and homes).

Table 3 provides a comparison between the proposed scheme and other methods used for the identification and classification of myocardial ischemia. It can be noted from Table 3 that many of the researchers employ a single heartbeat or that a small frame consists of a few consecutive beats. These schemes are differentiated by the beat type at high accuracy by using various analysis methods based on the time or/and frequency domain. However, these results do not reflect the physician’s interpretation, where the use of long records is necessary for a reliable interpretation. Moreover, single-beat characterization may cause uncertainty during the medical process—for example, what is the decision for a 2 min record that contains (after using single-beat analysis) 50% normal and 50% abnormal beats? Besides this, the most developed methods in Table 3 use a single database, which means that they need an extra process to adopt new data. The proposed automated scheme in this study employs the potential band region for both ST and PR segments. This procedure gives flexibility when using multiple databases and data sources, which assists fully automated-based interpretation systems. Comparing with S.G. Al-Kindi [52], L.N. Sharma [18], VK Sudarshan [22], and Kamal Jafarian who use a smaller frame, the proposed scheme shows an improvement in accuracy, sensitivity, and specificity. Additionally, JH Tan et al. [20] propose a new method that uses an eight-layer stacked CNN-LSTM with a blindfold to find the MI cases. However, it is based on using a single ECG lead, which can limit the ability to provide a complete interpretation, as many heart cases require more than a single lead to elucidate the situation. On the other hand, P. Barmpoutis et al. [6] conducted a study based on employing Grassmannian and Euclidean features and mapping them into a Hilbert space to improve the accuracy; this can be used with a single ECG beat, leading to incomplete interpretation and misleading even experts.

Instead of beats or frames, the use of extended intervals for ECG signal analysis has served to increase the precision and reliability of the proposed scheme and thus improve the overall performance of the automated procedure. The use of time frequency makes this scheme more insensitive to the noise source than other approaches that use time-based algorithms, in addition to reducing the need for long periods of interpretation of physical ECG data. The automated properties of this device can be expanded for use in homes where, in potential cases, it can provide a robust, quick, and non-invasive primary assessment.

The main advantages offered by our proposed scheme are summarized below:It comes with a high level of accuracy and sensitivity for the detection of MI cases.It excludes the need for ECG morphological detection.Its automated feature selection and extraction are combined in a single module.Its ten-fold cross validation ensures reliability and robustness.It does not require intricate computational machinery.The drawbacks of our scheme are listed as follows:Its detection of MI cases bypasses other heart problems such as arrhythmia.Due to the complexity of time time-frequency calculations, it is currently performed on computers. More efforts should be made to render the proposed scheme available on mobile phones.

## 5. Conclusions

Myocardial ischemic is the leading cause of death among CVD-afflicted patients. In view of this dilemma, we propose a new automated myocardial ischemia detection scheme based on the Choi-Williams distribution and multi-class SVM. In the context of classification, this scheme attained a level of 99.09% for accuracy, 99.49% for sensitivity, and 98.44% for specificity with the use of ST and PR extracted features. As such, the proposed scheme can be considered reliable and accurate for quick and early ischemia detection in settings that include hospitals, medical clinics, and even homes. Furthermore, this scheme holds an edge over other conventional heart diagnosis approaches, as it is non-invasive, cost-effective, and fast. We are of the opinion that efforts should be in the pipeline to extend the applicability of this scheme for the detection of other heart diseases.

## Figures and Tables

**Figure 1 sensors-21-02311-f001:**
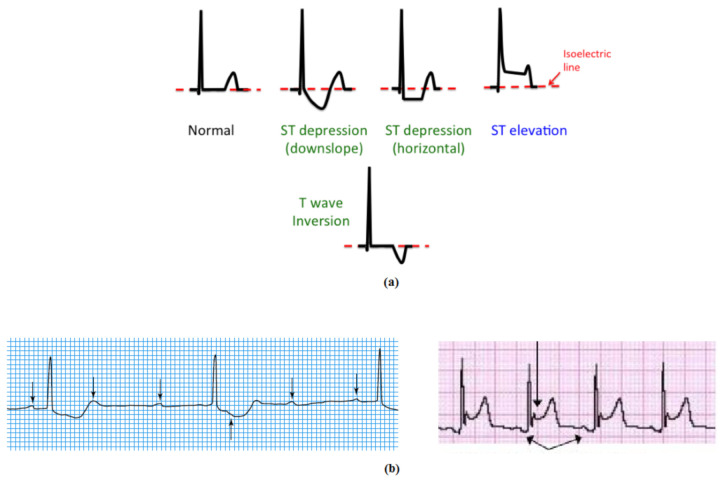
ECG morphology for (**a**) normal and ST elevation and (**b**) PR elevation [24].

**Figure 2 sensors-21-02311-f002:**
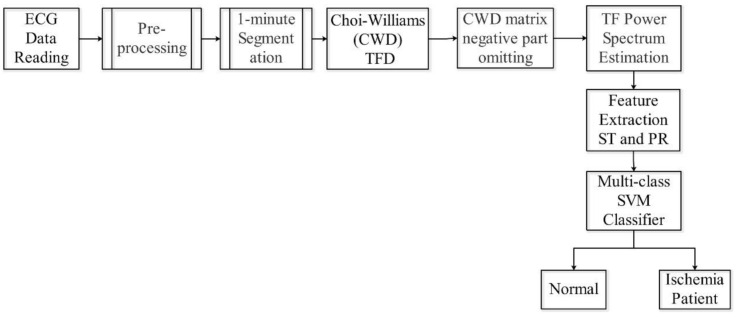
Proposed scheme.

**Figure 3 sensors-21-02311-f003:**
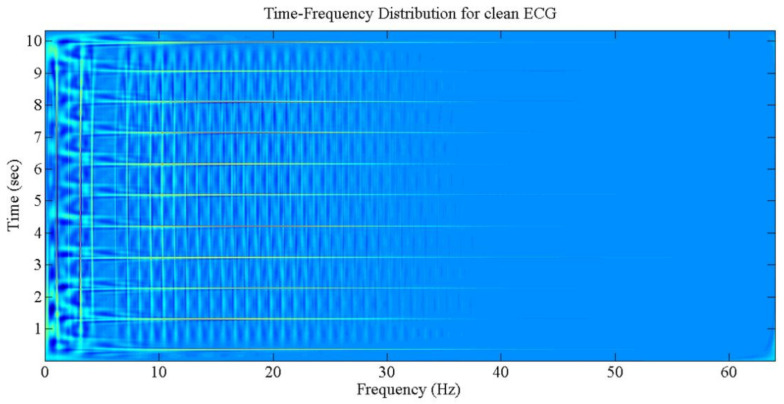
The TFD for normal and clean ECG.

**Figure 4 sensors-21-02311-f004:**
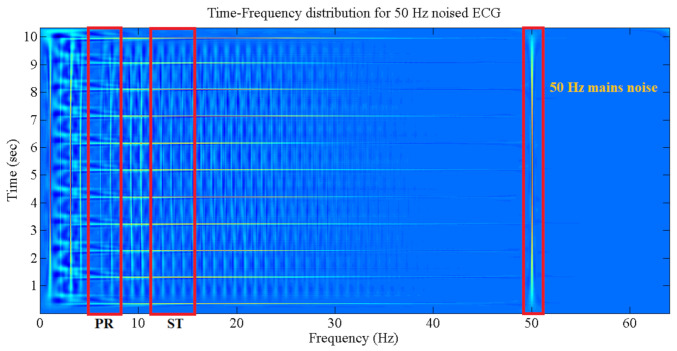
ECG frequency segments with regard to different sections.

**Figure 5 sensors-21-02311-f005:**
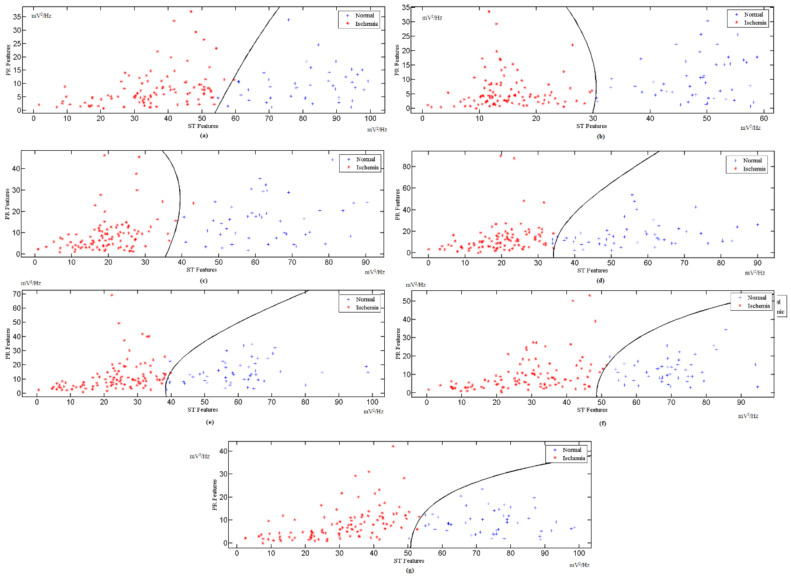
Indicator factor (F) SVM calculation for: (**a**) Lead I, (**b**) Lead III, (**c**) Lead V1, (**d**) Lead V2, (**e**) Lead V3, (**f**) Lead V4, and (**g**) Lead V5.

**Figure 6 sensors-21-02311-f006:**
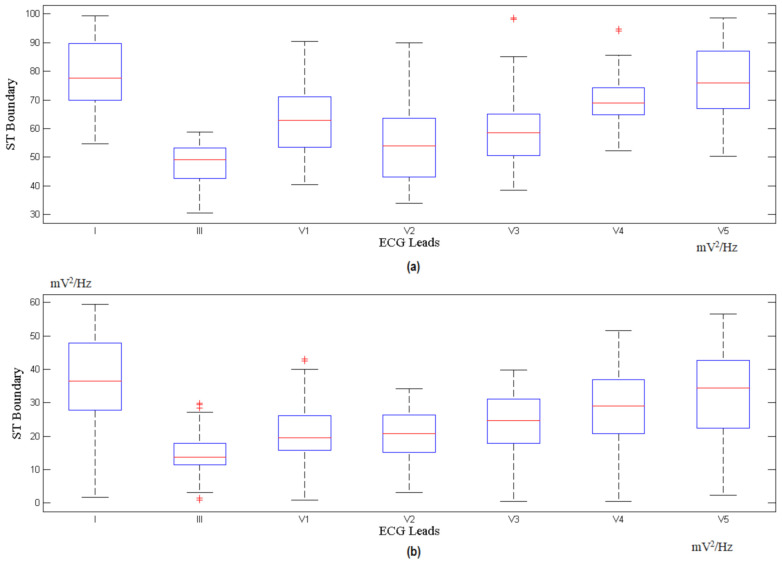
ST calculated boundary for (**a**) normal and (**b**) myocardial ischemia.

**Figure 7 sensors-21-02311-f007:**
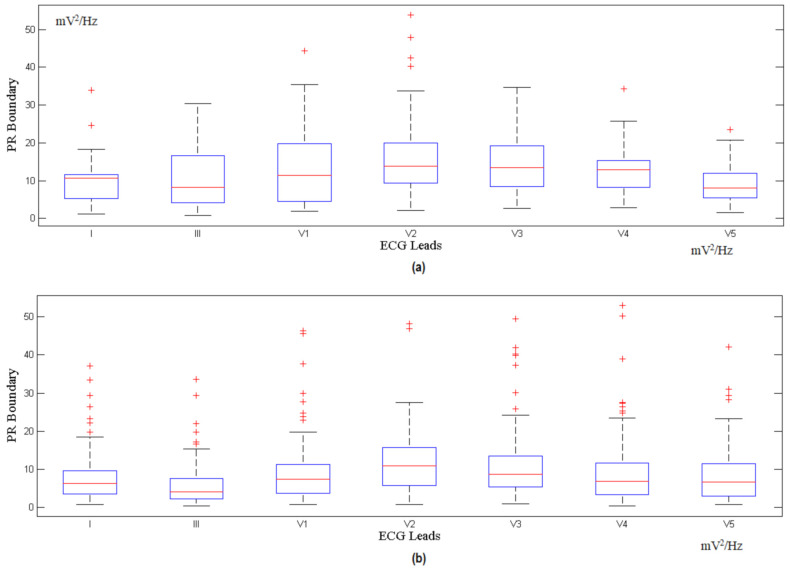
PR calculated boundary for (**a**) normal and (**b**) myocardial ischemia.

**Figure 8 sensors-21-02311-f008:**
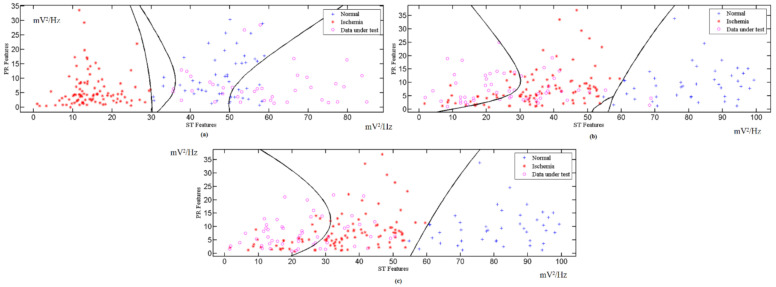
Testing results for: (**a**) Lead III normal case; (**b**) Lead I euro ST-T ischemic database; (**c**) Lead I collected ischemic data.

**Table 1 sensors-21-02311-t001:** Databases used for this study.

Database	Number of Subjects	Number of Records	Length (min) Per Record	Leads	Sampling Frequency	Total Used Data Length (min)/Lead	Training Used Data 80%	Validating Used Data 10%	Testing Used Data 10%
European ST-T database	78	88	120	2	250	21,600	17,280	2160	2160
Fantasia Normal database	40	40	120	1	250	4800	3840	480	480
Collected data from IBN-AL-NAFEES Hospital	30	30	30	7	133	180	144	18	18

**Table 2 sensors-21-02311-t002:** Testing results of different databases.

Database	Accuracy (%)	Sensitivity (%)	Specificity (%)
European ST-T database	99.31	99.33	96.31
Fantasia Normal database	99.19	99.49	99.61
Collected data from IBN-AL-NAFEES Hospital	98.78	99.65	99.42
Total	99.09	99.49	98.44

**Table 3 sensors-21-02311-t003:** Comparison between this work and state-of-the-art MI detection methods.

Method	Accuracy %	Sensitivity %	Specificity %	Database/ Remark	Subjects/ Records Number	Approach
Murugan [53]	NA	92.30	94.30	European ST-T/Beat	90 records	Ant-Miner algorithm
Jinhopark [54]	NA	95.70	95.30	European ST-T/Beat	90 records	Kernel density estimation (DWT based) with SVM
Safdarian [14]	94.00	NA	NA	PTB/Beat	290 subjects	T wave integral with KNN, PNN and ANN
B. Liu [16]	94.40	NA	NA	PTB/Beat	52 Normal 148 MI	ECG polynomial fitting algorithm PolyFit-based ECG
Jian Wang [48]	89.00	91.70	81.50	Hospital data collection	167 patients	Deep learning-based scheme
L. Sun [13]	NA	91.00	85.00	PTB/Beat	52 Normal 238 MI	ST segment, Polynomial fitting with KNN
Acharya U.R. [3]	98.50	99.70	98.50	PTB/Beat	52 Normal 148 CAD	DCT features based
Murthy [17]	90.51	96.19	NA	European ST-T/Beat	16 MI	Statistical analysis with PCA and SVM
M. Arif [9]	98.30	97.00	99.60	PTB/Beat	52 Normal 148 MI	KNN, Time domain feature extraction
J.H. Tan [20]	99.85	99.84	99.85	Fantasia, PTB Single lead	52 Normal 238 MI	8-layers stacked CNN-LSTM with Blindfold
P. Barmpoutis [6]	99.70	-	-	PTB/Beat	290 subjects	mapping of Grassmannian and Euclidean features into a Hilbert space
V.K. Sudarshan [22]	99.86	99.78	99.94	MIT-BIH Normal, Fantasia, and BIDMC/2-s Frame	73 subjects	Dual tree complex WT coefficients features with KNN
W.S. Kim [49]	NA	84.60	91.50	Collected data/HRV	20 Normal 64 Patients	HRV time and frequency measurements
E.S. Jayachan-dran [50]	95.00	NA	NA	MIT-BIH/Beat	6 Normal 2 MI	Time domain analysis
S.G. Al-Kindi [52]	93.70	85.00	100.00	PTB/ST-segments	20 Normal 20 MI	ST segment analysis by DWT
L.N. Sharma [18]	96.00	93.00	99.00	PTB/Frame	52 Normal 238 MI	ST segment analysed by DWT, KNN and SVM
Kamal Jafarian [55]	98.43	98.50	98.37	PTB/ST-segments	52 Normal 148 MI	CNN scheme used with DWT and PCA based features
This work	99.09	99.49	98.44	European ST-T, Fantasia, and Collected data/minute	92 Normal 266 MI	PR and ST feature extraction by using Choi-Williams and classified by Multi-Class SVM

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
