# Peer review of "An Automated High-Accuracy Detection Scheme for Myocardial Ischemia Based on Multi-Lead Long-Interval ECG and Choi-Williams Time-Frequency Analysis Incorporating a Multi-Class SVM Classifier"

_sensors, 2021, doi:10.3390/s21072311_

Round 1
Reviewer 1 Report
Although the topic of this work is of great interest, addressing the development of algorithms that help in the detection of CVD pathologies with high prevalence and mortality in the world the article in its current state cannot be considered for publication in the journal.
On the one hand, it is not clear whether, although the majority of studies in the literature focus on the extraction of beat-by-beat characteristics of the ECG signal, there is some work that also addresses the study of complete sections of the ECG signal and the advantage or improvement the present study entails in this regard. Another of the main problems lies in the actual writing of the document, which requires an exhaustive revision since there are totally unintelligible sentences and paragraphs that require a complete rewriting.
Regarding methods section
Clearer and more exhaustive explanations are required, as well as the analysis of the influence on the results of certain parameters. For example, line 120 indicates:
“Next, digital filters are conducted to remove ECG artifact that affected the quality of collected ECG, a 0.5 Hz high-pass FIR filter is carried out to remove baseline wander shifting noise, as well as 35 Hz to eliminate muscle noise artifacts [21- 2. 3]."
What type of filter is applied at 35 Hz? bandwidth?
A complete scheme of the multiclass classifier developed is necessary, including from the percentage of records and patients in each database used for training, validation and testing, as well as the derivations used in each of them and the criteria for decision-making when there can be more than one tap feeding the classifier inputs. It should include the ensemble of the binary classifiers used, as well as the methodology followed for the optimization of the SVM hyperparameters of each of them. Indeed this las issue has not been explained at all in the manuscript.
On the other hand, Has it been proven that the Choi-Williams distribution is the one with the best performance for the computation of the input characteristics to the classifier?
How have the interferences and artifacts been eliminated from the records? By visual inspection? Automatically? How has the signals been scaled? How has the segmentation been carried out in 1-minute sections?, what criteria was used?
It is not clear at a methodological level how the ST and PR segments have been mapped and how the boundaries were obtained. It is advisable to explain it with the support of a diagram and / or concrete examples.
Discussion
In the discussion section, much of what was commented in the introduction is repeated and the advantages, differences and similarities with other works published in the research field, both invasive and non-invasive, are not discussed in detail and the comparison table included is not enough.
Another questions:
To emphasize the need for a complete revision of the text. For instance:
- When formulas are included, all variables must be explained later.
- Figure 1 needs to indicate if the source is your own
- Figure 2 quite lacking. In general the captions are not sufficiently explanatory
- After formula 17 there is another formula that is not numbered.
- In line 214: 'num' is not found in the formulas, check.
- The units of Hz are missing in the summations of formula (17) and in which there is no number.
- Complete rewrite of point 3.1
- In figure 3 the axes have no units
- In Figures 4 and 5 the y-axis has no units
Author Response
Kindly, the comments response as attached

Reviewer 2 Report
The authors in the paper entitled: An Automated High Accuracy Detection Scheme for Myocardial Ischemia Based on Multi-Lead Long Interval ECG and Choi-Williams Time-Frequency Analysis Incorporating Multi-Class SVM Classifier, propose a new method for myocardial ischemia detection. The topic is interesting and the method sound promising however, I have a few comments with regards to this manuscript.
-First of all, the authors need to discuss (in the Introduction part) the limitations of other methods and what are the advantages of the proposed method against the already developed works. Furthermore, the authors need to add some missing works (for example the following work: Multi-lead ECG signal analysis for myocardial infarction detection and localization through the mapping of Grassmannian and Euclidean features into a common Hilbert space).
-Furthermore, in the 2.5 subsection, the authors need to state clear if the (Part one) indicator factor F computing is the training procedure of the proposed method.
-Moreover, I would like to ask the authors if the Table 2 presents the results of the data that already used for training? If yes, they have to remove this table and the most figures of this part and they have to present the result that are based on a test dataset.
-One more comment, the Tables 3,4,5 must be moved in the supplementary material section and I suggest the authors to add the results and ablation analysis for the 4 datasets that they tested their method.
Finally, I would suggest the authors to move the Figure 6 and the Table 7 to the discussion part and to extend their discussion, discussing the advantages and limitations of their method.
Author Response

(The authors gave the same response as above.)

Round 2
Reviewer 1 Report
Although some issues has been successfully addressed, other remains pending
Regarding methods section
A complete scheme of the multiclass classifier developed is necessary, including from the percentage of records and patients in each database used for training, validation and testing, as well as the derivations used in each of them and the criteria for decision-making when there can be more than one tap feeding the classifier inputs. It should include the ensemble of the binary classifiers used, as well as the methodology followed for the optimization of the SVM hyperparameters of each of them.
Discussion
Although the discussion section has been improved, it still requires justifying certain aspects in greater depth. For example the comparison shown in table 4 should be discussed in detail in the text, trying to justify the differences / similarities with respect to the different published works..
Another questions:
- Results in table 3 correspond to training, validation or test? This information must be included in the caption. Idem table 4.
- Review text layout
Author Response
The review response as attached.
Regards

Reviewer 2 Report
The authors improved the paper, however it still needs modifications in order to be published. First of all, with regards to the table that presents the results on the training dataset this needs to be removed. Also, the reader would want to see the test-validation results on Result Section (and not on training dataset). Then, the authors need to add all the methods discussed in the Introduction part, in Table 4. Furthermore, as the authors mention that they tested the proposed method in four datasets they need to present the results for the four datasets in Table 3. Moreover, ablation analysis would help readers to understand the method. Finally, it is necessary for the authors to proof read the main text as well as the figure captions.
Author Response

(The authors gave the same response as above.)

Round 3
Reviewer 1 Report
Reviewers' concerns has been addressed resulting in the corresponding changes in the manuscript.
The reviewed manuscript format, with change control enabled, had made the document review somewhat difficult. Please make a thorough review of the layout and numbering of the figures and tables, as well as the use of english in the text added.
Author Response
The authors sincerely thank the editors and reviewers for their comments and efforts to improving our manuscript. Please find below the response to the comments.
According to reviewer suggestion, an English language and style corrections are achieved.

Reviewer 2 Report
The manuscript has been improved and I only have some minor comments.
Initially, the authors mention that:
Line 107: Four ECG databases.
Table 1: The authors describe Five databases.
Table 2: The authors present results for 3 databases.
So, I would suggest the authors to make more clear how many databases have used and to present the results for the databases that they have used. If they have tested their algorithm in 3 databases they have to modify the Table 1 and the text in line 107.
Furthermore, in Table 3, I would suggest the authors to mention all the methods discussed in the Introduction part. For example the method: "Multi-lead ECG signal analysis for myocardial infarction detection and localization through the mapping of Grassmannian and Euclidean features into a common Hilbert space" and to discuss their limitations.
Author Response
The authors sincerely thank the editors and reviewers for their comments and efforts to improving our manuscript. Please find below the response to the comments.
Line 107: Four ECG databases. Table 1: The authors describe Five databases. Table 2: The authors present results for 3 databases. So, I would suggest the authors to make more clear how many databases have used and to present the results for the databases that they have used. If they have tested their algorithm in 3 databases they have to modify the Table 1 and the text in line 107. |
As the reviewer’s suggestion, the number of used data has been modified in table 1 and line 107 |
Furthermore, in Table 3, I would suggest the authors to mention all the methods discussed in the Introduction part. For example the method: "Multi-lead ECG signal analysis for myocardial infarction detection and localization through the mapping of Grassmannian and Euclidean features into a common Hilbert space" and to discuss their limitations. |
As the reviewer’s suggestion, table 3 is updated with the methods discussed in the introduction. |
